

# The effect of transcutaneous electrical nerve stimulation on fatigue recovery in recreational rowers

Yinan Zhou[1],[*], Xinyu Mao[2],[*], Wen Luo[2], Kaiyuan Qu[2], Anqi Lu[2], Jiayu Zang[2], Yu Chen[2], Rui Wu[3] and Dan Wang[2]

[1] Department of Physical Education, Tongji University, Shanghai, China
[2] School of Athletic Performance, Shanghai University of Sport, Shanghai, China
[3] School of Electrical & Electronic Engineering, University College Dublin, Dublin, Ireland
[*] These authors contributed equally to this work.

Corresponding author
Dan Wang, wangdan@sus.edu.cn

## ABSTRACT

**Background:** The quadriceps muscle group plays an important role in rowing performance but is also susceptible to fatigue. Transcutaneous electrical nerve stimulation (TENS) has been shown to be effective in delaying the onset of fatigue, suggesting it may play a role in fatigue recovery after exercise. This study aimed to assess the effects of TENS on quadriceps recovery following a 2000-m rowing ergometer exercise.

**Methods:** Twenty-eight recreational rowers were randomly allocated in a TENS group and a sham-TENS group. Following a 2000-m rowing ergometer exercise, the TENS group immediately received a 20-min TENS intervention, while the sham-TENS group received a placebo TENS. Ratings of perceived exertion, blood lactate (BLa), heart rate, the peak torque of knee extensors, and the surface electromyography (sEMG) activity (vastus lateralis and rectus femoris) of both limbs were recorded before and immediately after the exercise, and after the intervention.

**Results:** The TENS group showed a superior recovery in muscle strength (increased 20.76% *vs* 9.44%, $F_{(2, 106)} = 11.549$, $p < 0.001$, $\eta^2_p = 0.179$) and BLa (reduced 7.93 *vs.* 6.34 mmol/L, $F_{(1, 26)} = 6.768$, $p = 0.015$, $\eta^2_p = 0.207$) when compared to sham-TENS. However, no time by group interaction was observed in EMG responses in both time ($F_{(2, 214)} = 1.268$, $p = 0.268$, $\eta^2_p = 0.012$) and frequency ($F_{(2,214)} = 0.997$, $p = 0.371$, $\eta^2_p = 0.009$) domains.

**Conclusion:** A TENS intervention may be effective in facilitating recovery from quadriceps muscle fatigue in recreational rowers. Thus, it could serve as an alternative recovery treatment to enhance fatigue recovery in recreational rowers.

## INTRODUCTION

Rowing is one of the classic Olympic sports that requires a high level of physical preparation (*Treff, Winkert & Steinacker, 2021*). The rowing cycle, which commences with knee and torso flexion (recovery phase), is subdivided into a drive phase and a recovery phase, with a time allocation ranging from 0.9 to 1.7 (*Soper & Hume, 2004*). The cycle culminates in knee and torso extension , along with upper extremity flexion, which marks
the drive phase (*Kabaciński et al., 2020*). Knee extensions are essential in rowing, as they significantly contribute to the propulsion of the rowing boat by pushing against the foot stretcher (*Husmann et al., 2016*). Rowers face sternward and generate force by transferring muscle power from the lower limbs to the upper limbs, which in turn leverages the oars and rowlocks to move the boat forward (*Chang et al., 2024*). Additionally, the quadriceps muscle group is thought to constitute the "driving" elements at the source of the power generated by elite rowers (*Penichet-Tomás & Pueo, 2017*). Trained male rowers possess greater quadriceps muscle volume, indicating their greater potential for performance (*Ema et al., 2013*). It has been observed that a 2000-m rowing ergometer exercise resulted in significant quadriceps muscle fatigue (*Husmann et al., 2017*).

Muscle fatigue is defined as an exercise-induced reduction in the force-generating capacity of a muscle (*Gandevia, 2001*). It is resulted from multiple mechanisms, including metabolic changes, such as ATP depletion, accumulation of $H^+$ and inorganic phosphate (Pi) (*Fitts, 1994*; *Allen, Lamb & Westerblad, 2008*), and reactive oxygen species generation (*Constantin-Teodosiu & Constantin, 2021*). These changes can disrupt muscle homeostasis, leading to decreased muscle performance and increased recovery time (*Allen, Lamb & Westerblad, 2008*). Postexercise recovery is crucial for maintaining high levels of exercise performance and reducing the risk of injury. Fast postexercise recovery helps restore physiological and psychological homeostasis, which is essential for athletes facing frequent competitions and high physical demands (*Bestwick-Stevenson et al., 2022*). Insufficient recovery may result in fatigue, underperformance, and an increased risk of injuries (*Bestwick-Stevenson et al., 2022*).

Transcutaneous electrical nerve stimulation (TENS) has been widely used in both medical and sports-related settings for rehabilitation, training, and recovery (*Babault et al., 2011*). It involves the use of adhesive electrodes placed on the skin surface to deliver pulsed electrical stimulation (*Babault et al., 2011*). *Astokorki & Mauger (2017)* reported that TENS can attenuate perceived exercise-induced pain in healthy populations and significantly improve endurance performance in both submaximal isometric single-limb exercise and whole-body dynamic exercise. Similarly, *Behm et al. (2019)* found that the unilateral application of TENS had a crossover or global effect on the perception of pain or discomfort prior to a submaximal isometric fatiguing contraction protocol. A 20-min TENS intervention prior to a fatiguing protocol has been proven to extend the time to knee extensors failure by inducing a global pain modulatory response (*Behm et al., 2019*). Indeed, several previous studies have explored the effects of TENS, and their findings imply its potential role in fatigue recovery. For example, *So, Ng & Ng (2007)* reported the transcutaneous electrical acupoint stimulation could improve quadriceps strength recovery within 15 min, suggesting pain control as a potential mechanism for fatigue recovery. *Degani et al. (2024)* demonstrated that TENS significantly reduced muscle soreness and improved recovery time following intense physical activity, likely due to its ability to enhance blood flow and reduce the accumulation of metabolic by-products. Therefore, it has been speculated that TENS may have the potential to aid fatigue recovery. However, there is limited evidence to support its effectiveness in improving the fatigue recovery of key physiological variables (*e.g.*, muscle strength and neuromuscular function)

or in maintaining exercise performance. Therefore, the aim of this study was to investigate the effect of TENS on fatigue recovery of the quadriceps muscles among recreational rowers after a 2000-mrowing ergometer exercise. It was hypothesized that TENS would be more effective in post-exercise recovery.

## MATERIALS AND METHODS

### Participants

Twenty-eight participants were recruited and randomly divided into a TENS group and a sham-TENS group using a computer-generated randomization program implemented in SPSS (Version 20; IBM Corp., Armonk, NY, USA). The inclusion criteria for the participants were as follows: (1) college recreational rowers (training 2–3 times a week for 2 h each time); (2) aged 19–24 years. The exclusion criteria were as follows: (1) participants with lower limb muscle or bone injuries in the past 3 months; (2) participants who had ingested caffeine, alcohol, or any sports drinks within 24 h before the testing. The general characteristics of the participants are included in Table 1. Written informed consent was obtained from all participants. The study was approved by the Ethical Committee of the Shanghai University of Sport before testing commenced (Reference No. 102772021RT081).

### Experimental design

The participants arrived at the laboratory on the experimental day 2 h after eating a meal. The familiarization was performed prior to the assessment but with a 20-min rest before the actual testing was executed, and this was mandatory for all participants. A single-blind experimental design was employed, with each trial involving only one participant. The participants were not aware of the specific sensations of the TENS intervention in this experiment prior to the trial. In the sham-TENS group, the placebo was administered with the device powered on but not activated. Therefore, the procedure was consistent for both groups. First, a standardized warm-up (consisting of running at 9 km/h for 5-min, followed by dynamic lower-limb stretching exercises (Silva et al., 2017)). The following data were collected before and immediately after a 2000-m rowing ergometer exercise and after a 20-min recovery intervention: (1) ratings of perceived exertion (RPE), (2) blood lactate (BLa), (3) peak torque (PT) and surface electromyography (sEMG) of the knee extensors for both limbs, and (4) heart rate (HR).

Fatiguing exercise was performed on a Concept II rowing ergometer (Model D; Concept II Inc., Rochester, NY, USA). Research has identified significant fatigue in the voluntary activation of knee extensors in athletes following a 2000-m rowing session (Husmann et al., 2017). Participants were asked to perform a 2000-m rowing ergometer exercise, which is the standard distance at the Olympic Games and the World Rowing Championships and successfully mimics competition-induced fatigue (Husmann et al., 2017). The Concept II rowing ergometer provides a highly realistic simulation of water-based rowing and is a staple in the training regimen of rowing athletes. The exercise began with the initiation of a paddle stroke with a resistance coefficient of four for females

**Table 1 Participants and baseline (T1) characteristics.**

|  | TENS group | Sham-TENS group | Significance |
|---|---|---|---|
| Participants ($n$) | 14 (11 males, 3 females) | 14 (11 males, 3 females) |  |
| Age (years) | 20.7 ± 1.6 | 21.6 ± 1.9 | $p = 0.193$ |
| Height (cm) | 174.1 ± 9.3 | 176.2 ± 6.8 | $p = 0.506$ |
| Body mass (kg) | 67.6 ± 10.0 | 71.5 ± 8.6 | $p = 0.278$ |
| BMI (kg/m$^2$) | 22.2 ± 1.9 | 22.9 ± 2.3 | $p = 0.328$ |
| 2000-m time (min) | 7.8 ± 0.9 | 7.2 ± 2.2 | $p = 0.412$ |
| Training experience (months) | 12.7 ± 8.5 | 13.1 ± 8.0 | $p = 0.910$ |
| Length of the right lower limb (cm) | 43.0 ± 3.1 | 43.4 ± 2.7 | $p = 0.680$ |
| Length of the left lower limb (cm) | 43.0 ± 3.0 | 43.3 ± 2.7 | $p = 0.793$ |
| RPE | 9.5 ± 2.0 | 9.6 ± 1.1 | $p = 0.907$ |
| HR (bpm) | 102.0 ± 9.6 | 103.7 ± 10.3 | $p = 0.778$ |
| BLa (mmol/L) | 2.9 ± 0.3 | 2.7 ± 0.6 | $p = 0.494$ |
| Peak Torque (Nm) | 91.5 ± 35.7 | 101.2 ± 32.5 | $p = 0.294$ |
| EMG RMS VL (mV) | 0.510 ± 0.242 | 0.411 ± 0.222 | $p = 0.107$ |
| EMG RMS RF (mV) | 0.368 ± 0.187 | 0.382 ± 0.336 | $p = 0.844$ |
| EMG MDF VL (Hz) | 88.4 ± 18.6 | 80.7 ± 22.0 | $p = 0.155$ |
| EMG MDF RF (Hz) | 75.0 ± 17.8 | 68.5 ± 15.0 | $p = 0.143$ |

and five for males, according to their respective training habits. Verbal encouragement was consistently provided throughout the entire process.

## Measurements
### Ratings of perceived exertion
Participants' RPE was assessed using the Borg 20-point scale to evaluate the level of perceived exertion (*Buemann & Tremblay, 1996*).

### Heart rate
HR was measured using a heart rate monitor (Polar heart rate strap; Polar Electro Oy, Kempele, Finland) (*Hettiarachchi et al., 2019*).

### Blood lactate
A 20 μL capillary blood sample was obtained *via* fingerpick from each participant's ring finger for BLa concentration analysis. The BLa concentrations were analyzed onsite using a lactate analyser (Biosen C-Line, EKF Diagnostics, Barleben, Germany).

### Peak torque of the knee extensor
Participants were instructed to sit on the edge of in a rigid chair with their knees and hips flexed at 90°. The right or left limb was randomly selected for testing, followed by the other limb. The researchers positioned the curved pad of a handheld dynamometer (MicroFET3; Hoggan Scientific LLC, Salt Lake City, UT, USA) vertically at the distal end of the lower limb (parallel to the lateral malleolus) to assess the maximal strength of both quadriceps muscles. The Hoggan handheld dynamometer has been known for its excellent reliability

(ICC = 0.90) and validity (ICC = 0.89) (*Mentiplay et al., 2015*). Instructions provided to participants for all trials were 'at the count of three, push as hard and as fast as you can and hold that contraction until researcher say relax'. Each test lasted five seconds and ended after a steady maximal force was produced by the participant. Constant verbal encouragement was provided throughout the testing. Each limb underwent two attempts with a 1-min break in between. An additional attempt was allowed when the coefficient of variation was higher than 5%. The results were multiplied by the lever arm (the distance between the lateral femoral epicondyle and the lateral malleolus) to calculate the PT. The highest recorded value was utilized for the analysis.

## Surface electromyography

sEMG was recorded from the vastus lateralis (VL) and the long head of the rectus femoris (RF) muscles during each PT test (Bagnoli$^{TM}$ Desktop System; Delsys, Natick, MA, USA). Prior to application of the electrode, the skin was shaved, scrubbed, wiped with alcohol pads and marked with a mark pen. Electrodes were placed on the muscle belly of the VL (2/3 of the line from the anterior superior iliac spina to the lateral side of the patella) and the RF (50% of the line from the anterior superior iliac spina to the superior part of the patella) according to the SENIAM guidelines (*Hermens et al., 1999*).

Surface EMG signals were sampled at 1,111 Hz and processed using custom-written programs (MATLAB R2023a; MathWorks, Natick, MA, USA). Surface EMG signals were first off-line band-pass filtered between 20 and 500 Hz using a fourth-order zero-lag Butterworth filter. To quantify the surface EMG time and frequency components at PT, a moving root mean square (RMS) was performed to detect the plateau of torque. Then, the RMS and median frequency (MDF) were calculated over a period of 1,000 ms corresponding to the previously identified plateau of torque. The EMG RMS and MDF of the VL and RF muscles recorded from PT after fatiguing exercise and recovery intervention, were normalized relative to their RMS and MDF values obtained from pre-fatigue PT.

## TENS intervention

TENS interventions can be achieved by utilizing various stimulation parameters, namely, frequency and intensity (*DeSantana et al., 2008*). These can be categorized into three types: conventional TENS (high-frequency, low-intensity), acupuncture-like TENS (low-frequency, high-intensity), and intense TENS (high-frequency, high-intensity) (*Mokhtari et al., 2020*). Conventional TENS has been shown to block the transmission of nociceptive afferent fibers in the spinal cord by stimulating large-diameter group II myelinated afferent fibers (*Mokhtari et al., 2020*). Acupuncture-like TENS stimulates both small diameter myelinated (Aδ) and unmyelinated (C) afferent nerves. This subsequently activates extra segmental descending pain inhibitory pathways, thereby producing a spatially diffuse analgesic effect (*Johnson, 2021*). Intense TENS aims to activate Aδ fibers and block harmful messages through extra-nodal analgesic mechanisms, similar to acupuncture-like TENS (*Johnson, 2021*). In addition, it has been demonstrated to produce rapid and significant analgesia both during and after stimulation (*Johnson, 2021*). Accordingly, the stimulus

variables selected for this study were high frequency and high intensity—specifically, 120 mA, 200 μs, and 150 Hz (*Mokhtari et al., 2020*).

The TENS pain management program of the Compex SP 2.0 (DJO Global Products Inc., Carlsbad, CA, USA) electrical stimulation trainer was utilized for recovery intervention in the TENS group for 20 min. Self-adhesive electrode pairs (2 pairs of 5 cm × 5 cm, 1 pair of 10 cm × 5 cm) were positioned on the VL (2/3 of the line from the anterior superior iliac spina to the lateral side of the patella), the vastus medialis (80% of the line between the anterior superior iliac spina and the joint space in front of the anterior border of the medial ligament), and the RF (50% of the line from the anterior superior iliac spina to the superior part of the patella) on both legs (*Arumugam et al., 2020*). Current intensity was determined according to the participant's maximum tolerance to the electrical stimulation (*Cuesta-Gómez et al., 2019*). Every 5 min, the researchers asked the participants whether the sensation was comfortable or decreasing and adjusted the intensity of TENS accordingly based on the feedback. The sham-TENS group underwent a 20-min placebo TENS intervention with electrodes placed on the skin but no electric current, and was told "This type of stimulation is supposed to help recovery by using a subthreshold stimulus that you will not able to perceive".

## Statistical analysis

Statistical analyses were conducted using IBM SPSS Statistics (Version 20.0; IBM Corp., Armonk, NY, USA). Normal distribution of the data was tested using the Kolmogorov–Smirnov test, showing that all variables were normally distributed. An analysis of variance (ANOVA) test with repeated measures was used to determine significant interactions between time [prefatigue (T1), postfatigue (T2), and recovery (T3)] and group (TENS and sham-TENS) on RPE, BLa, PT of knee extensors, and sEMG activities. A two-way ANOVA with repeated measures was performed with two between-subject factors (muscle and group) and one within-subject factor (time: T1, T2 and T3) on EMG MDF and RMS. When a significant interaction effect was detected, *post hoc* analysis was performed for further analysis. When there was no interaction effect, a simple effects analysis was performed. Effect size was calculated as partial eta squared. Statistical significance was set at $p < 0.05$. Data are reported as means ± SD within the text and displayed as means ± SEM in the figures.

# RESULTS

The baseline (pre-fatigue, T1) data are summarized in Table 1. No significant difference between the two groups was observed in any of the anthropometric and baseline measures.

## RPE

There was no significant interaction between time and group in RPE ($F_{(2, 52)} = 1.069$, $p = 0.350$, $\eta^2_p = 0.039$). However, a main effect of time was observed for RPE scores, showing a similar increase for both groups at T2 (TENS:17.8 ± 1.1 *vs.* sham-TENS: 18.5 ± 1.5, $p < 0.001$) and a decrease ($p < 0.001$) at T3 (TENS: 9.7 ± 2.2 *vs.* sham-TENS: 9.4 ± 1.8).

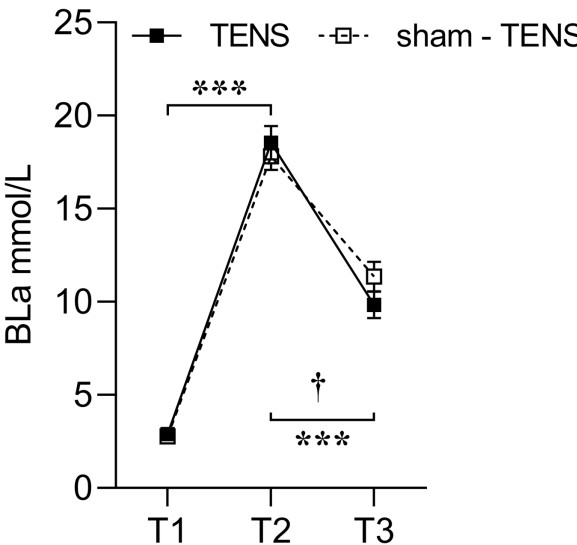

**Figure 1 Blood lactate changes in the TENS group and the sham-TENS group.** T1 before the 2000-m race; T2 immediately after the 2000-m race; T3 after the 20-min intervention; T1 *vs.* T2, T2 *vs.* T3, ∗∗∗$p < 0.001$; interaction effect between T2 and T3, † $< 0.001$.

### HR

Only a main effect of time was observed for HR ($F_{(2, 52)} = 794.355$, $p < 0.001$, $\eta^2_p = 0.968$), showing a similar HR increase immediately at T2 ($p < 0.001$) (TENS: 184.1 ± 9.2 bpm *vs.* sham-TENS: 180.4 ± 5.5 bpm) and a decrease at T3 ($p < 0.001$) for both groups (TENS: 102.7 ± 13.1 bpm *vs.* sham-TENS: 102.9 ± 10.1 bpm).

### BLa

A significant interaction was observed in BLa between the two groups from T2 to T3 ($F_{(1, 26)} = 6.768$, $p = 0.015$, $\eta^2_p = 0.207$), showing that the TENS group had a greater decrease (17.76 to 9.83 mmol/L) compared with the sham-TENS (17.70 to 11.36 mmol/L) (Fig. 1).

### PT of knee extensors

A significant group-by-time interaction was observed in terms of PT ($F_{(2, 106)} = 11.549$, $p < 0.001$, $\eta^2_p = 0.179$). *Post hoc* testing revealed a significant difference in PT between the two groups from T2 to T3 ($F_{(1,53)} = 17.378$, $p < 0.001$, $\eta^2_p = 0.247$) but not between T1 and T2 ($p = 0.131$). PT showed a similar reduction from T1 to T2 in both groups (TENS: 87.6 ± 9.4 Nm *vs.* sham-TENS: 82.8 ± 13.6 Nm; $p < 0.001$) but a different increase from T2 to T3; a greater increase/recovery was observed in TENS (108.4 ± 13.9 Nm, $p < 0.001$) compared to sham-TENS (91.7 ± 16.0 Nm, $p < 0.01$) group (Fig. 2).

### sEMG

#### RMS

No time by group interaction was observed in EMG RMS ($F_{(2, 214)} = 1.268$, $p = 0.268$, $\eta^2_p = 0.012$). Only a time main effect was observed in RMS ($F_{(2, 206)} = 5.797$, $p = 0.004$,

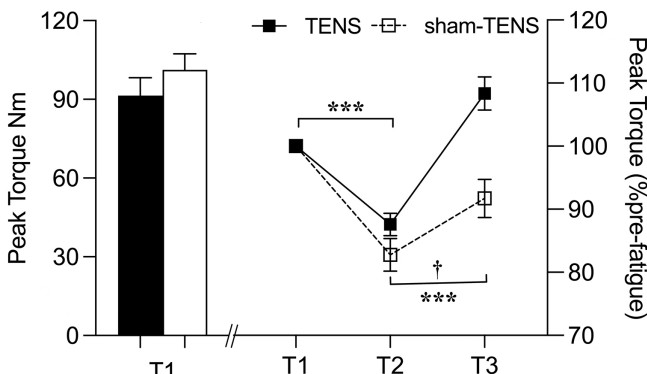

**Figure 2 Peak torque changes in the TENS group and the sham-TENS group.** T1 before the 2000-m race; T2 immediately after the 2000-m race; T3 after the 20-min intervention, T1 *vs.* T2, T2 *vs.* T3, ***$p < 0.001$; interaction effect between T2 and T3, † < 0.001.

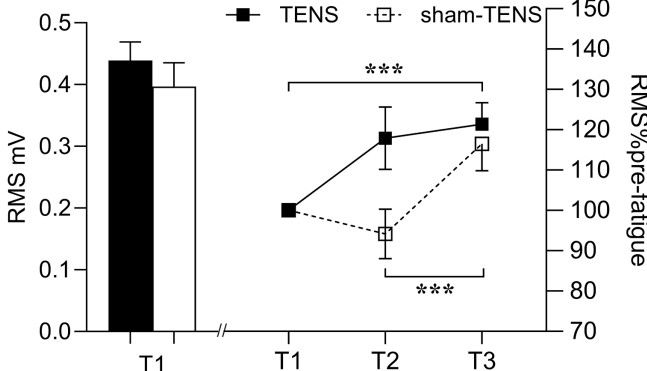

**Figure 3 RMS changes in the TENS group and the sham-TENS group.** T1 before the 2000-m race; T2 immediately after the 2000-m race; T3 after the 20-min intervention, T1 *vs.* T2, and T2 *vs.* T3; ***$p < 0.001$.

$\eta^2_p = 0.053$), showing that RMS at T3 was greater than at either T2 ($p < 0.001$) or T1 ($p < 0.001$) in both groups (Fig. 3).

### MDF

There was no significant interaction between time and group ($F_{(2,214)} = 0.997$, $p = 0.371$, $\eta^2_p = 0.009$) in MDF. However, an interaction effect between time and muscle ($F_{(2,214)} = 9.560$, $p < 0.001$, $\eta^2_p = 0.082$) was observed, showing that the VL did not change throughout the testing, but the RF decreased at T2 ($p < 0.001$) then increased from T2 to T3 ($p < 0.001$) (Fig. 4). Moreover, at T1, the MDF of the RF was greater than that of the VL ($84.62 \pm 20.58$ *vs.* $71.80 \pm 16.68$ Hz; $F_{(1,107)} = 13.685$, $p < 0.001$, $\eta^2_p = 0.113$) (Fig. 4).

## DISCUSSION

The aim of this study was to evaluate the effect of TENS on the recovery of athletic performance following a 2000-m rowing race. In line with our hypothesis, both the TENS

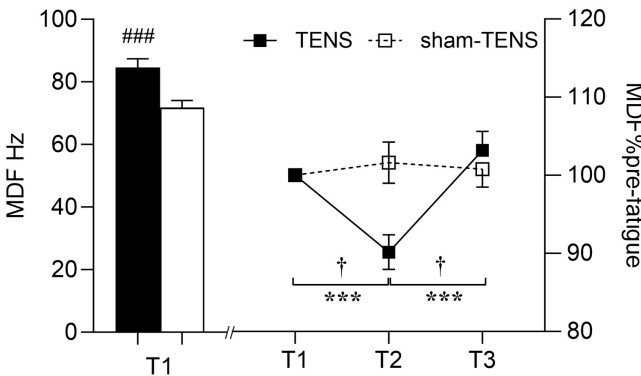

**Figure 4 MDF changes in the VL and RF muscles.** T1 before the 2000-m race; T2 immediately after the 2000-m race; T3 after the 20-min intervention, the VL *vs* RF at T1 (baseline), ###$p < 0.001$; T1 *vs*. T2, T2 *vs*. T3, ∗∗∗$p < 0.001$; interaction effect between T2 and T3, † $p < 0.001$.

and the sham-TENS groups exhibited recovery after a 20-min recovery period, as reflected by improvements in RPE, HR, BLa, PT, and sEMG activity. This overall recovery may be attributable to the natural recovery process. Notably, significant intervention effects were observed in PT and BLa, indicating that the TENS group recovered significantly superior than the sham-TENS group.

## RPE and HR

The 2000-m rowing ergometer exercise resulted in a similar increase in RPE and HR between the TENS and sham-TENS groups. After the 2000-m rowing ergometer exercise on the Concept II rowing ergometer, RPE surged to 17 or higher, while the immediate HR of both groups was over 180 bpm. These results demonstrated that comparable fatigue was successfully induced in both groups. RPE is a commonly employed subjective measure of exertion during physical activity, demonstrating correlations with a range of physiological metrics, including muscular force, heart rate, ventilation, respiratory rate, oxygen uptake, and BLa concentrations (*Tucker, 2009*). *Borg (1982)* affirmed that RPE serves as the foremost indicator of physical strain, integrating signals from peripheral working muscles and joints as well as from the central cardiovascular system, respiratory functions, and nervous system. Moreover, RPE is comprised of not only physiological factors but also psychological and affective components, alongside other mediators (*Coquart et al., 2012*). Therefore, the overall sensation of exertion experienced during exercise can be considered the conscious verbal expression of the integration of physiological and psychological factors (*Tucker, 2009*). The competence for rating perceived effort is typically evaluated by correlating RPE with objective measures, such as HR (*Reinke & Schmitz, 2023*). A high correlation between HR and RPE has been reported (*Reinke & Schmitz, 2023*). Therefore, RPE and HR changes proved that successful fatigue was induced by the 2000-m rowing ergometer exercise in both the TENS and sham-TENS groups.

The TENS intervention did not result in any significantly different alterations in RPE and HR between the TENS and sham-TENS groups following the 20-min intervention.
However, previous findings on the effect of TENS on RPE and HR changes after fatigue were not consistent with the results of the present study. *Cheung & Jones (2007)* found that TENS significantly reduced perceived exertion during physical activity. The participants reported lower RPE scores when TENS was applied, indicating that TENS can help alleviate the perception of effort during exercise. Since the Borg scale index is used to evaluate subjective fatigue, it may be difficult to compare only these ratings between subjects, as perceptions of fatigue vary greatly for each individual (*Al-Mulla, Sepulveda & Colley, 2011*). *Haddad et al. (2013)* suggested that there may be additional psychobiological factors beyond fatigue, stress, delayed onset of muscle soreness and sleep that influence RPE during exercise. Indeed, TENS has been shown to significantly accelerated the recovery of HR after treadmill running, with participants who received TENS showing a faster reduction in HR compared to those who did not receive the treatment (*Cheung & Jones, 2007*). Psychological factors can also influence HR. For example, emotional states, such as anger, fear, and dread, have been demonstrated to significantly affect HR (*Al-Mulla, Sepulveda & Colley, 2011*). In this case, the sham-TENS group received a placebo intervention and performed the same operation as the other group, with members of both groups having electrodes placed on their legs. This produced a psychological placebo effect, which may have contributed to the lack of a significant difference in RPE and HR between the two groups after the intervention.

## BLa

Following the 2000-m rowing ergometer exercise, both the TENS and sham-TENS groups exhibited a comparable rapid increase in BLa levels (exceeding 18 mmol/L) (84.32% *vs*. 84.34%). This phenomenon can be attributed to the fact that lactate is produced at a faster rate than the body's capacity to remove it, resulting in its accumulation during high-intensity exercise (*Johnson, 2021*). Furthermore, the accumulation of lactate lowers the pH of the blood, thereby exacerbating muscle fatigue (*Flotyńska et al., 2021*). Therefore, it is important to ensure the timely reduce of lactate to maintain a high level of competitive performance (*Hinojosa, Hearon & Kowalsky, 2021*).

Compared to the sham-TENS group, the TENS group exhibited a much higher lactate removal rate (47.03% *vs*. 36.25%) after the recovery phase. After maximal intensity exercise, active recovery removes accumulated lactic acid from the blood at a faster rate than passive recovery, thereby facilitating the release of lactic acid produced by the muscles for uptake by the exercising and nonexercising muscles (*Bangsbo et al., 1993*). TENS has been reported to stimulate damaged frontal nerve cell tissue and enhance local blood circulation (*Preetam et al., 2024*). Additionally, TENS-induced vasodilation improves oxygen delivery and facilitates more effective blood lactate reduce (*Naik et al., 1999*). The lactate is oxidized and used as fuel by the muscles, heart, and brain (*Proia et al., 2016*). It has been demonstrated that electrical stimulation is effective in increasing the return of venous blood to the heart, thereby increasing cardiac output (*Grunovas et al., 2007*). Therefore, compared to natural recovery from fatigue, TENS intervention offers significant advantages.

## PT of knee extensors

After completing the 2000-m rowing ergometer exercise, the TENS and sham-TENS groups experienced a similar decline (12.41% *vs.* 17.22%) in PT of both limbs ([Fig. 2]). Muscle fatigue primarily stems from diminished neural control over muscle activities, thereby reducing force-generating and joint motion capability (*Liu et al., 2021*). Furthermore, the 2000-m ergometer exercise on the Concept II rowing ergometer likely prompted a substantial glycolytic response among athletes, possibly leading to excessive $H^+$ accumulation and subsequent cellular pH reduction (*Lancha Junior et al., 2015*). This acidic environment may disrupt myosin and actin interactions, consequently impairing athletes' strength and performance quality.

Compared to the sham-TENS group, the TENS group exhibited a superior recovery (23.7% *vs.* 10.49%) in PT of the knee extensors of both limbs. This finding confirms our hypothesis and may be attributed to several factors. First, the direct activation of muscle fibers by TENS increases muscle activity levels, thus increasing muscle strength after the intervention (*Sartori et al., 2024*). Second, TENS enhances neural adaptation by increasing motor unit recruitment and firing rates to provide additional neural stimulation, potentially increasing muscle strength (*Bax, Staes & Verhagen, 2005*). Third, TENS modulates cardiovascular function by acting on the autonomic nervous system, indirectly improving circulation and increasing blood flow to the skin, which helps accelerate blood flow and recovery from postexercise fatigue (*Tomasi et al., 2015*). Fourth, TENS reduces pain by activating the body's inhibitory mechanisms to reduce the excitability of the central nervous system (*Dailey et al., 2013*); this effect is achieved by stimulating large afferent fibers that, in turn activating downstream inhibitory systems, thereby reducing pain sensations (*Dailey et al., 2013*). Meanwhile, TENS stimulates the release of endogenous opioid peptides (*e.g.*, endorphins), which reduce pain perception (*Astokorki & Mauger, 2017*). Another potential mechanism is that TENS may play a role in reducing peripheral sensitization of muscle afferent nerves, a process that may occur after intense exercise and contribute to delayed onset muscle soreness. In addition, TENS may influence both peripheral and central fatigue, as well as blood flow to the muscle (*Zambolin et al., 2024*). Therefore, TENS may facilitate fatigue recovery and enhance muscle strength by increasing muscle activation and neural adaptation, modulating cardiovascular function and reducing pain and peripheral sensitization of muscle afferent nerves (*Johnson et al., 2022*).

## Surface EMG

The sEMG analysis revealed changes in the quadriceps muscles of both groups following fatigue without time or group interaction. RMS is employed in the analysis of muscle activity, with a particular focus on the evaluation of muscle fatigue (*Rampichini et al., 2020*). It represents a prevalent metric for assessing the characteristics of sEMG time-domain signals, quantifying signal power in a manner that reflects alterations in signal amplitude (*Del Vecchio et al., 2017*). This approach offers a means of evaluating muscle activation patterns (*Del Vecchio et al., 2017*). The MDF is an index used in studies

of spectral shifts and can be defined as "the frequency which divides the power spectrum in two parts with equal areas" (*Al-Mulla, Sepulveda & Colley, 2011*). A reduction in this MDF is frequently associated with muscle fatigue, as fatigue results in a shift in the frequency component of the EMG signal toward lower frequencies (*Husmann et al., 2017*). As muscle strength declines and fatigue sets in, the time-domain features of sEMG (RMS) rise, while the frequency-domain features (MDF) decline (*Luttmann, Jger & Laurig, 2000*).

No significant time or group interaction was observed in RMS following the recovery. A study by *Behm et al. (2019)* found that pre-exercise TENS significantly reduced RMS in the biceps femoris, which may be related to the delayed onset of fatigue. The main reason for the inconsistent discoveries may be explained by the different intervention timing of TENS. However, the reason why TENS applied before and after fatigue produced these controversial results remains unclear. It is speculated that variations in the intensity, duration and frequency of TENS might contribute to its effect on fatigue recovery. Further investigation is needed to valid this. However, RMS increased similarly following the 20-min recovery between both groups. The interference EMG signals can only provide a more 'global' view on the EMG responses in the time and frequency and therefore do not necessarily truly reflect motor unit (MU) recruitment strategies and individual MU properties (*Keenan et al., 2006*; *Del Vecchio et al., 2017*). Although the similar increase at T3 was observed in EMG RMS between two groups, the underlying neurophysiological mechanisms may differ. In the TENS group, this increase could be attributed to the enhanced muscle activation and pain-reducing effects of TENS (*Tousignant-Laflamme et al., 2017*). Indeed, *Chan et al. (1999)* have reported that TENS can increase the fatigue resistance of the larger and more fatigable MUs, potentially leading to an increase in overall EMG amplitude. On the other hand, in the Sham-TENS group, the increase in EMG amplitude may have resulted from natural recovery, in which the MUs with smaller amplitudes were recruited (*e.g.*, slow-twitch MUs) (*Jensen, Pilegaard & Sjøgaard, 2000*). This suggests that, in sham-TENS group, more but small MUs may have been activated to achieve the same increase in EMG amplitude as observed in TENS group. In addition, an interaction effect between time and muscle observed in EMG MDF showing that VL did not change throughout the test, but the RF decreased after the fatigue exercise and then increased after the recovery phase. This is consistent with the findings of previous study (*Mullany et al., 2002*). First, it may be attributed to potential differences in muscle fiber type between the RF and the VL muscles (*Wu et al., 2019*). The proportion of type II muscle fibres in RF is greater than in VL muscles, which are less fatigue-resistant and more easily fatigued during longer and higher-loaded contractions (*Mathur, Eng & MacIntyre, 2005*). This could lead to a decrease in the post-fatigue MDF. Second, in rowing (*i.e.*, multi-articular movements), the RF, as a bi-articular muscle, may recruit more motor units at an earlier stage during knee extension than a mono-articular biarticular muscle (*i.e.*, VL), resulting in greater fatigue (*Mathur, Eng & MacIntyre, 2005*). This distinct recruitment strategy may result in a more pronounced decline in the MDF of the RF following fatigue. Furthermore, research has demonstrated that the MDF performance of the RF and VL may vary in response to different contraction intensities (*Vøllestad, 1997*).

Specifically, the MDF decline in the RF may be more pronounced during high-load (80% MVC) contractions, while the difference may be less evident during low-load (20% MVC) contractions (*Mathur, Eng & MacIntyre, 2005*). Additionally, the observation that the MDF of the RF declines after fatigue and then increases after recovery may be linked to the physiological recovery process of the muscle following fatigue. This suggests that the RF may undergo a rebound in its EMG activity characteristics after a period of recovery, while the VL may not exhibit this apparent dynamic change (*Vøllestad, 1997*). In conclusion, these factors contribute to the differences in the sEMG amplitudes between the RF and VL under specific conditions.

### Limitation

The present study comes with some limitations. Firstly, the study adopted a medium effect size (0.25) to determine the sample size, resulting in a total requirement of 28 participants. This approach may lack methodological rigor. Secondly, *Green, Parro & Gabriel (2014)* reported a minimum of 10, but no more than 15 contractions were necessary to constitute a familiarization session of isometric dorsiflexion contractions. In this study, only one familiarization session was implemented before testing. Future studies should incorporate adequate familiarization procedures. Thirdly, blood lactate was assessed from capillary blood taken from the fingertip. *Feliu et al. (1999)* reported that blood lactate concentrations collected from fingertip and earlobe could be different. Future research should consider collecting the blood sample from the earlobe for blood lactate analysis. The fourth limitation is that we had to temporarily remove the EMG electrodes during the TENS recovery intervention for the placement of the TENS electrodes. Removing and reattaching EMG electrodes likely adds unexplained variance, although the location of the EMG electrodes was marked on the skin to ensure the consistency. Future investigations should systematically address this concern to minimize potential data discrepancies induced by EMG electrode detachment. Lastly, this study focused exclusively on the immediate effects of TENS interventions on fatigue recovery. Future research could explore the long-term effects of TENS on fatigue recovery as well as its application in other muscle groups.

## CONCLUSIONS

In summary, this study suggested that TENS may effectively aid in the fatigue recovery of the quadriceps in recreational rowers. Therefore, it could serve as an assistive strategy for fatigue recovery in this population.

### Funding

This research was funded by a grant from the Program for Overseas High-level Talents at Shanghai Institutions of Higher Learning under Grant No. TP2019072, and Shanghai Key Lab of Human Performance (Shanghai University of Sport) under Grant No.

11DZ2261100. The funders had no role in study design, data collection and analysis, decision to publish, or preparation of the manuscript.

### Grant Disclosures
The following grant information was disclosed by the authors:
Shanghai Institutions of Higher Learning: TP2019072.
Shanghai Key Lab of Human Performance (Shanghai University of Sport): 11DZ2261100.

### Competing Interests
The authors declare that they have no competing interests.

### Author Contributions
- Yinan Zhou performed the experiments, authored or reviewed drafts of the article, recruit the participants, and approved the final draft.
- Xinyu Mao conceived and designed the experiments, performed the experiments, analyzed the data, prepared figures and/or tables, authored or reviewed drafts of the article, and approved the final draft.
- Wen Luo performed the experiments, prepared figures and/or tables, authored or reviewed drafts of the article, and approved the final draft.
- Kaiyuan Qu performed the experiments, prepared figures and/or tables, and approved the final draft.
- Anqi Lu performed the experiments, prepared figures and/or tables, and approved the final draft.
- Jiayu Zang performed the experiments, authored or reviewed drafts of the article, and approved the final draft.
- Yu Chen performed the experiments, authored or reviewed drafts of the article, and approved the final draft.
- Rui Wu analyzed the data, prepared figures and/or tables, authored or reviewed drafts of the article, and approved the final draft.
- Dan Wang conceived and designed the experiments, authored or reviewed drafts of the article, and approved the final draft.

### Human Ethics
The following information was supplied relating to ethical approvals (*i.e.*, approving body and any reference numbers):
Ethical Committee of the Shanghai University of Sport (Reference No. 102772021RT081).

### Data Availability
The raw data for TENS study is available in the Supplemental File.

## Supplemental Information

Supplemental information for this article can be found online at http://dx.doi.org/10.7717/peerj.19388#supplemental-information.

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
