# Peer review of "The effect of transcutaneous electrical nerve stimulation on fatigue recovery in recreational rowers"

_PeerJ, doi:10.7717/peerj.19388_

## Round 0.1 · original submission · Major Revisions

Dear Authors

Thank you for submitting your work to PeerJ. Kindly address all the concerns raised by the referees and submit the updated manuscript for further review.

Reviewer 1 ·

Basic reporting

no comment

Experimental design

The randomized controlled design is valid. However, given the baseline differences between groups and the acute design of the study, a crossover-design might have improved study power.

Validity of the findings

no comments

Additional comments

Dear editorial board, dear authors,
thank you for the opportunity to review this interesting manuscript. This study investigated the effect of TENS as an acute recovery tool. Although this study may focus on an interesting aspect, there are a few major concerns about this manuscript. Therefore, in its current form, this manuscript is not suited for publication. Please see specific comments below.

L81: Why did you choose to provide only a post-hoc power-analysis and not also a priori sample size estimation? Moreover, why didn’t you choose a crossover-design?
L111: When was RPE assessed?
L114: Polar will typically record HR data at a rate of 1 to 10Hz. How was the data processed?
L117: Why did you choose the fingertip and not earlobe? Typically, blood lactate concentration in rowing is assessed from capillary blood obtained from the earlobe. Using the fingertip will lead to substantially higher lactate values than typically assessed at the earlobe (https://link.springer.com/article/10.1007/BF03655586). Moreover, when exactly were these samples obtained?
L121: How did you ensure that participants could replicate the exact same position at all different time points? And how was the handheld dynamometer fixed in a standardized position? Could you provide reliability data for this measurement device?
L124: Did you assess both legs simultaneously?
L127: So familiarization was performed immediately prior to the assessment? Was this mandatory? Do you think that there might have been some learning effects – especially when considering the ~10-20% higher torque data at T3 compared to T1?
L130: However, this is not the real arm. The real arm would be the distance between the lateral femoral epicondyle and the pad of the dynamometer.
L131: Why did you only use the highest value, when you obtained more than one measurement? Using an average value will provide a more sensitive measure. How do your results change, if you use the average value of both trials?
L165: This suggests that you had to remove the EMG electrodes. How did you ensure that the electrodes were placed at the same position after the recovery period?
L175: I understand that it is hard to provide a blinded sham-intervention when applying electrical stimulation and your approach is absolutely reasonable and very common. However, did you also ask participants after the intervention whether they believed this or not?
L187: For all results in general: I recommend including information on the df of the F-Statistic and a more detailed description of post-hoc results (i.e., more information than just p-values). Moreover, I recommend also including standard deviations for all results.
L201: You may want to think about whether this information on main effects is relevant given you found a statically significant interaction effect.
L207: I recommend providing absolute values, as the percentage change is already provided as a figure.
L233: I recommend changing this sentence as you did only find no statistically significant differences in RPE, which should not be confused with similar results – let alone similar fatigue.
L255: Wouldn’t you agree that therefore a crossover-design would have been better suited?
L269 & L277: I’m not quite sure what these percentage values represent. Also they should be reported with SD.
L280: So why would you think didn’t you find any condition differences in lactate at T3?
L297: I’d recommend changing the wording of “faster recovery” as you only had two time points (immediately post and 20 minutes post) and no time course and considerable baseline differences.
L299: Why do you think that a further activation of the fatigued muscle fibres did result in a superior recovery?
L314/319: As mentioned earlier, you most likely had to remove the EMG electrodes and reattach them later. How do you think did this influence the EMG data?
L335: Why do you think there are two different explanations for the different conditions even though you got the same result?
L368: The limitation section should be amended, and methodological limitations of the study should be discussed here.

Figure 1 and 2: You may want to think about not including these figures into the manuscript as they (i) could be easily reported within the text, (ii) do not show any relevant differences between both conditions (it is quite hard to distinguish the two different conditions from each other), (iii) they do not show any variance in data for figure 2 and most likely not the SD but rather the SEM for figure 1 (please check?), and (iv) the indication of statically different values between time points via brackets makes it hard to associate them with any condition.
Figure 3: please indicate whether this is mean values ± SD or SEM?
Figure 4: for TENS it seems that peak torque was substantially higher at T3 compared to T1. Do you think this might be a result of learning effects – especially when considering the substantially lower PT in TENS at baseline. Also, again, please indicate whether this is mean values ± SD or SEM?
Figure 4 and 5: there seems to be a substantial baseline difference between the two conditions?
Figure 6: the title of the figure seems odd, as it is not possible to distinguish between TENS and sham in this figure. Moreover, the muscle*time interaction was not reported in the statistics section and I also wonder, what the hypothesis behind this analysis would be? Could you please clarify?
Table 1: I recommend including information on the performance level (i.e., 2000-m time or power) to allow for a better comparison. Moreover, you may want to think about deleting the last column (table 1 fallacy). Also, a rather small detail: weight should be changed to mass.

Reviewer 2 ·

Basic reporting

The article is very well written and clear. The motivations of the article are clearly articulated.

Experimental design

The authors mention that a "computer generated randomization program" was used, but this wording isn't clear. Was the program computer-generated? It seems that the randomization was carried out by a program. I would clarify this wording.

The authors mentioned that the experiments occurred 2h after a meal. This is a great consideration that the authors made, was this always a full meal? Carbohydrate content can influence fatigue.

Verbal encouragement was provided. Were the experimenters blinded to the experimental condition of the participant being tested?

The sampling rate is a bit unusual for EMG (1111Hz). Is this a typo? All of the systems that I have used around this rate sample at either 1000 or 1024? Granted, I do not have experience with the Bagnoli system.

Validity of the findings

No comments. The methods are appropriate and the findings are well communicated with no concerns related to over-analysis of the collected data.

Additional comments

This is a very well-communicated and motivated study. Some clarity could be added to the methods where noted above.

Reviewer 3 ·

Basic reporting

Clear and unambiguous, professional English used throughout. We suggested to improve the language throughout and check for spelling errors

Experimental design

Assessment of peak torque using hand-held dynamometer can raise question on the reliability of the measures. We suggested the authors to share the data of their familiarisation and relevant literature to justify the use of this procedure to assess strength.

Validity of the findings

Impact and novelty not assessed. Meaningful replication encouraged where rationale & benefit to literature is clearly stated.

Annotated reviews are not available for download in order to protect the identity of reviewers who chose to remain anonymous.

---

## Round 0.2 · Minor Revisions

Dear Authors

Although the referees are positive about your paper one of the reviewers has some some more concerns, kindly make the necessary changes and submit with a rebuttal letter.

Reviewer 1 ·

Basic reporting

no comment

Experimental design

no comment

Validity of the findings

no comment

Additional comments

Dear editorial board, dear authors,
I highly appreciate your revised manuscript! Although the manuscript improved, I have some minor comments regarding the responses to my initial points. Please see specific comments below:


Page 3 Line 81:
Why did you choose to provide only a post-hoc power-analysis and not also a priori sample size estimation? Moreover, why didn’t you choose a crossover-design?
Response: Thank you for the question. No previous studies have used TENS for recovery intervention following a fatiguing exercise. Prior to the study, we adopted a medium effect size (0.25) to determine the sample size, resulting in a total requirement of 28 participants. However, considering this approach potentially lacked rigor, we performed a post-hoc power analysis based on our experimental data, which indicated a power of 0.99. Besides, we believe that study design of randomized controlled trail (RCT), which compare the effects of interventions through random grouping, also could be used in our study. Many previous studies with fatigue recovery intervention have employed the RCT design. Please see our previous answer to this question.

REPLY: Thank you for this clarification. However, I feel that this approach is flawed, as post hoc power analyses based on observed data are considered redundant with the p-value and do not provide any new insights. Since post hoc power is directly related to the observed effect size, this analysis is tautologic. Moreover, the inclusion of post-hoc power analyses might be misleading, as a non-significant result with low power might suggest a sample size issue, but a non-significant result with high post hoc power falsely implies the study was adequately powered despite failing to detect an effect. Therefore, I recommend removing the post-hoc power analysis data all together and add this to the limitation section.


Page 9 Line 127
So familiarization was performed immediately prior to the assessment? Was this mandatory? Do you think that there might have been some learning effects – especially when considering the ~10-20% higher torque data at T3 compared to T1?
Response: Yes, the familiarization was performed prior to the assessment but with a 20-min rest before the actual testing was executed, and this was mandatory for all participants. While previous studies utilized a 10-minute rest period to prevent fatigue after familiarization (Sanchez-Gonzalez et al., 2011; Guilkey et al., 2022), the present study has extended this interval to 20 minutes. This methodological refinement minimized potential fatigue-induced confounding effects on testing outcomes while preserving experimental rigor. We have clarified it in the manuscript (lines 128-130).
Moreover, participants were asked to perform maximal contraction at least twice at their maximal effort (peak torque). Two trials ensuring less than 5% variation for PT demonstrated similar performance, indicating that a ‘true’ PT was obtained for each participant. An additional attempt was required if the variation between the first two trials was greater than 5%. We have included it in the manuscript (lines 182-183). Moreover, peak torque at T3 in the TENS group increased by about 10% compared to them at T1, but there was no significant difference between these two time points statistically. Therefore, we believe there is no learning effect in this test.
REPLY: Thank you for this thorough reply. A single familiarization trial just before testing is insufficient for reliable values (https://doi.org/10.1139/apnm-2013-0253). I recommend noting this in the limitations section.

Page 15 Line 314/319
As mentioned earlier, you most likely had to remove the EMG electrodes and reattach them later. How do you think did this influence the EMG data?
Response: Thank you for your question. Yes, we had to temporarily remove the EMG electrodes during the TENS recovery intervention in order to place the TENS electrodes. However, to ensure the EMG electrodes were placed on the same location immediately after the TENS intervention, the location of the EMG electrodes was marked on the skin with a marker pen prior to the testing, ensuring the consistency. This has been included in the manuscript (line 190-191). With similar procedures, a previous study showed a good inter-day reliability (ICC = 0.72) for EMG amplitudes recorded from the vastus lateralis muscle in males and females (Wu et al., 2016). Cicchetti (1994) gives the following often quoted guidelines for interpretation ICC inter-rater agreement measures: less than 0.40-poor, between 0.40 and 0.59-fair, between 0.60 and 0.74-good, between 0.75 and 1.00-excellent (Cicchetti, 1994). Therefore, we believe the influence of the removal and reattaching of the EMG electrodes on the EMG data was acceptable.
REPLY: Thank you for your response. Although an ICC of 0.72 suggests good reliability, the 95% CI should also be considered (doi: 10.1016/j.jcm.2016.02.012). Also, higher variability in a dataset (e.g., including both sexes) can inflate ICC values (DOI: 10.1519/15184.1). Removing and reattaching electrodes likely adds unexplained variance. I recommend discussing this in the article’s limitations section.

Reviewer 3 ·

Basic reporting

The manuscripts has improved considerably following reviewers comments. no further comments from my side

Experimental design

The manuscripts has improved considerably following reviewers comments. no further comments from my side

Validity of the findings

The manuscripts has improved considerably following reviewers comments. no further comments from my side

Additional comments

The manuscripts has improved considerably following reviewers comments. no further comments from my side

---

## Round 0.3 · accepted · Accept

Dear Authors thank you for addressing all the comments of the reviewers.

Reviewer 1 ·

Basic reporting

no comment

Experimental design

no comment

Validity of the findings

no comment